# Population pharmacokinetic modeling of missed mycophenolate mofetil doses: Impact on exposure and exploration of mitigation strategies

Franck Maizaud[1], Selim Arraki-Zava[1], Hamza Sayadi[1], Yeleen Fromage[1,2], Pierre Marquet[1,2,3], Jean-Baptiste Woillard[1,2,3], Caroline Monchaud[1,2,3]*

1 Department of Pharmacology, Toxicology and Pharmacovigilance, CHU Limoges, Limoges, France, 2 INSERM UMR-1248 "Pharmacology & Transplantation", Limoges, France, 3 Fédération Hospitalo-Universitaire "Survival Optimization in Organ Transplantation" (FHU SUPORT), Limoges, France

* caroline.monchaud@inserm.fr

## Abstract

### Background

Missed doses of mycophenolate mofetil (MMF) are frequent in transplant recipients and may lead to subtherapeutic exposure to mycophenolic acid (MPA), potentially compromising graft function. However, current guidance on how to manage such deviations remains empirical and is not supported by pharmacokinetic evidence.

### Methods

We used two validated population pharmacokinetic models of MPA to simulate steady-state exposure in virtual cohorts of renal transplant recipients treated with MMF. Scenarios included full missed doses, delays in intake (2–10 hours), and compensation strategies (50% or 100% added at the next dose). Simulations were performed for six dose regimens (500–1250 mg BID) and analyzed in terms of $AUC_{0-12h}$, proportion of patients outside the target range (30–60 mg·h/L), and time to return to steady state.

### Results

A fully missed dose resulted in a 40–60% reduction in $AUC_{0-12h}$, with delayed return to steady state (up to 72 hours). Delays ≤6 hours caused minimal impact (<15% reduction). Full-dose compensation frequently overshot the therapeutic window, especially in higher-dose regimens. In contrast, adding 50% of the missed dose restored exposure safely and rapidly. When exact compensation was not feasible, rounding down to the nearest available formulation minimized overexposure.

**Data availability statement:** All raw data are within the manuscript and its Supporting Information files.

**Funding:** The author(s) received no specific funding for this work.

**Competing interests:** I have read the journal's policy and the authors of this manuscript have the following competing interests: FM, SA-Z, HS, and YF declare no competing interests. CM has received speaker honoraria and/or research funding from Chiesi, Astellas, Pfizer, and Gilead. PM has received speaker and consulting honoraria and/or research grants from MedinCell, Chiesi, Sandoz, Astellas, and Bristol Myers Squibb (BMS). JBW has received speaker honoraria from Chiesi, Astellas, Pfizer, Vifor, and Gilead. This does not alter our adherence to PLOS ONE policies on sharing data and materials.

**Abbreviations:** $AUC_{0-12h}$, Area under the concentration–time curve from 0 to 12 hours; BID, Bis in die (twice daily); $C_0$, Trough concentration; CL/F, Apparent clearance after oral dosing; CrCl, Creatinine clearance; IPV, Intra-patient variability; IRB, Institutional Review Board; MMF, Mycophenolate mofetil; MPA, Mycophenolic acid; MPAG, Mycophenolic acid glucuronide; PK, Pharmacokinetics; PopPK, Population pharmacokinetics; Q/F, Apparent intercompartmental clearance; TDM, Therapeutic drug monitoring; Tlag, Lag time before absorption; V1/F, Apparent central volume of distribution; V2/F, Apparent peripheral volume of distribution.

## Conclusions

Model-based simulations suggest that a rational response to MMF dose omissions can mitigate underexposure while avoiding toxicity. These strategies may support clinicians in managing missed doses and optimizing immunosuppressive therapy in kidney transplant recipients.

## 1 Introduction

Mycophenolate mofetil (MMF) is a cornerstone of immunosuppressive therapy after solid organ transplantation. Although marketed as a fixed-dose drug, it exhibits high interindividual pharmacokinetic (PK) variability. The poor correlation between the administered dose, trough concentrations ($C_0$), and the area under the curve (AUC) of its active metabolite, mycophenolic acid (MPA), has been well documented [1]. Several randomized trials, such as APOMYGRE [2], have demonstrated improved outcomes with AUC-guided MMF dose adjustment. As AUC better reflects systemic exposure, therapeutic drug monitoring (TDM) based on the $AUC_{0-12h}$ is recommended, particularly during the first year post-transplant, with a target range of 30–60 mg·h/L [1,3,4]. While no studies have clearly established the optimal exposure beyond the first year, it is generally assumed that maintaining AUC values within this range remains clinically relevant. Nonetheless, fixed-dose strategies are still widely used in clinical practice, and TDM remains underutilized.

Non-adherence, as defined by the World Health Organization, includes missed doses, timing deviations, or both, and is a prevalent challenge in transplantation [5,6]. In kidney transplantation, poor adherence is reported in 15–40% of adults and up to 60% of adolescents, with increased prevalence beyond two years post-transplant [7–9]. This phenomenon is also common among liver transplant recipients, where it often manifests as irregular follow-up or missed clinic appointments and is associated with late acute rejection and graft loss [10]. Importantly, even minor deviations from prescribed regimens can lead to adverse outcomes, including late acute rejection and graft loss. Indeed, non-adherence is estimated to contribute to approximately 20% of late rejection episodes and 16% of graft losses [11–15].

Maintaining adherence to immunosuppressive therapy is therefore essential to ensure optimal drug exposure, preserve graft function and contribute to long-term graft survival [7,8,16,17]. Various tools have been developed to assess adherence, including self-report questionnaires (e.g., BAASIS, MARS-10), pill counts, electronic monitoring (e.g., MEMS), and mobile health applications [7,18,19]. Additionnaly, intra-patient variability (IPV) in tacrolimus concentrations is increasingly used as a surrogate marker for adherence, with high IPV being associated with poorer outcomes, though results remain inconsistent [20–23].

To improve adherence, various strategies have been proposed, including regimen simplification, behavioral interventions and educational strategies, digital reminders via smartphone apps, and financial support [24–26]. Nevertheless, in long-term therapies such as MMF, occasional missed doses are likely to occur even among adherent

patients. Despite this, practical and evidence-based recommendations on how to manage missed doses are lacking. Patient information leaflets recommend taking the missed MMF dose as soon as remembered, without doubling the next dose. However, this advice can be confusing if the omission is discovered shortly before the next scheduled intake, and it is not based on published pharmacokinetic data.

To date, the pharmacokinetic impact of missed doses and dose recovery strategies has been explored for tacrolimus, but not for MMF. Although clinicians have expressed a need for such information, no study has specifically evaluated the impact of missed MMF doses. Given the narrow therapeutic index of MMF and the clinical risks associated with MPA underexposure, addressing this knowledge gap is clinically relevant and timely. The present study therefore aimed to evaluate the impact of a missed or delayed MMF dose on MPA exposure using population PK modeling, and to explore recovery strategies that could mitigate the associated loss of exposure.

## 2  Materials and methods

As this work was based exclusively on pharmacokinetic simulations and did not involve any patient data or biological samples, ethical approval and informed consent were not required. The code used to perform the simulations and analyses is provided in the Supplemental Data for full transparency and reproducibility.

### 2.1  Pharmacokinetic model selection

A literature search was conducted on PubMed using the following keywords: pharmacokinetic model, MMF, renal transplantation (Fig 1).

Two population pharmacokinetic (PopPK) models of MPA were selected for simulation. The first model, published by Rong et al. [27], was developed from 27 stable adult kidney transplant recipients co-treated with immediate-release tacrolimus. The second model, by van Hest et al. [28], was based on data from 140 renal transplant recipients receiving ciclosporin.

Both models were implemented in R (version 4.3.2) using the mrgsolve package and were validated by comparing simulated $AUC_{0-12h}$ values with those reported in the original publications. Key pharmacokinetic parameters and model characteristics are summarized in Table 1. For the purpose of these simulations, residual variability was fixed at a minimal value ($\sigma = 0.001$) to isolate the influence of inter-individual variability and covariates on exposure.

### 2.2  Simulation scenarios

Steady-state MPA pharmacokinetic profiles were simulated for six virtual patient groups (n = 1,000 each): three dose levels per model (500, 750, and 1000 mg BID for Rong's model; 750, 1000, and 1250 mg BID for van Hest's model). Extreme $AUC_{0-12h}$ values ($0.1^{st}$ and $99.9^{th}$ percentiles) were excluded from the analysis [29].

Each group was subjected to the following scenarios: (i) Missed dose: complete omission of one dose, resulting in a 24-hour gap between administrations; (ii) Delayed dose: delays of 2, 4, 6, 8, or 10 hours, producing intervals of 14–22 hours before and 2–10 hours after the delayed intake; (iii) Recovery strategies: following a missed dose, an additional 50% or 100% of the missed dose was added at the next scheduled intake. When 50% doses were not available, existing commercial formulations (250 mg and 500 mg capsules) were used to approximate recovery doses.

### 2.3  Outcome measures

The primary outcome was the relative difference in $AUC_{0-12h}$ compared to steady-state exposure. For each patient, $AUC_{0-12h}$ was computed for the 12-hour interval corresponding to the missed, delayed, or compensated dose. The proportion of individuals outside the therapeutic target range (30–60 mg·h/L) was estimated.

Return to steady state was defined as the first 12-hour dosing interval during which $AUC_{0-12h}$ was within ±10% of the baseline steady-state value. Acceptability criteria for the scenarios were as follows: the delayed dose was considered

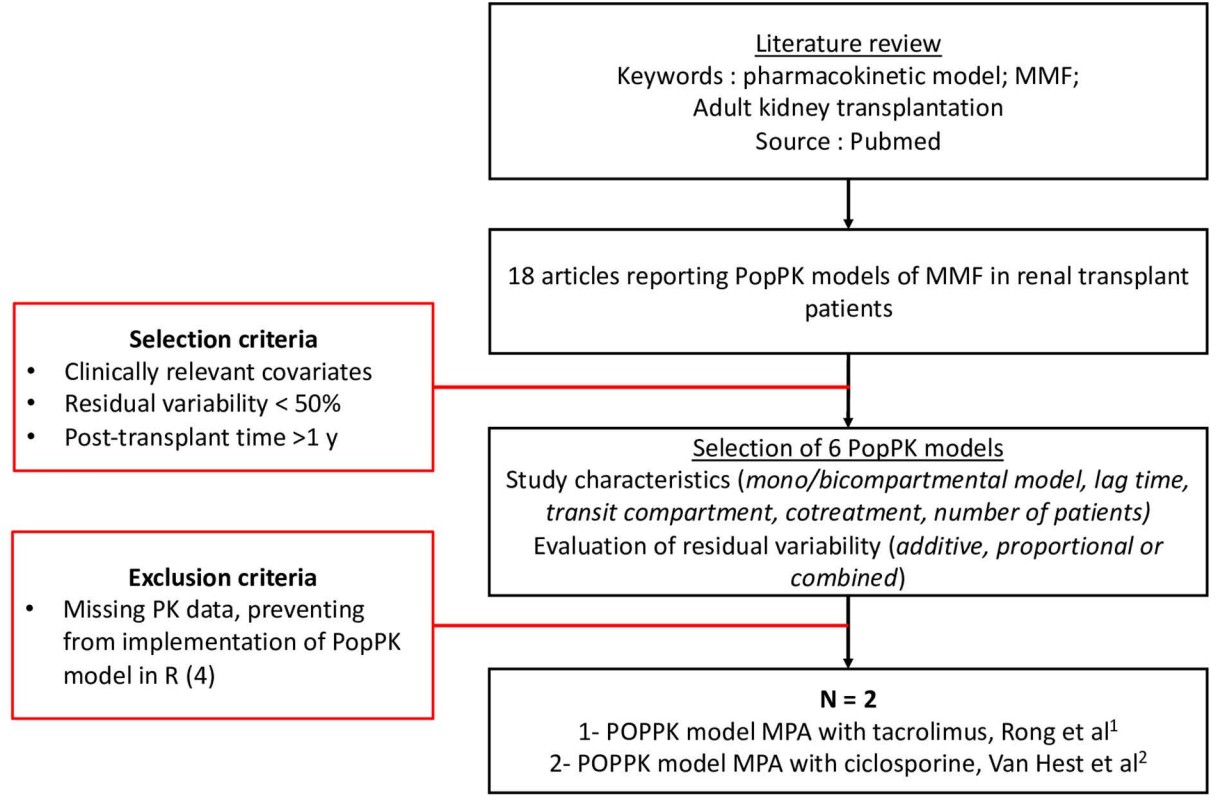

[1]Rong Y, Mayo P, Ensom MHH, Kiang TKL. Population Pharmacokinetics of Mycophenolic Acid Co-Administered with Tacrolimus in Corticosteroid-Free Adult Kidney Transplant Patients. Clin Pharmacokinet. 2019 Nov;58(11):1483–95.
[2]van Hest RM, van Gelder T, Vulto AG, Mathot RAA. Population pharmacokinetics of mycophenolic acid in renal transplant recipients. Clin Pharmacokinet. 2005;44(10):1083–96.

**Fig 1. Flowchart of bibliographic review and model selection for population pharmacokinetic analysis.**

acceptable if mean $AUC_{0-12h}$ decrease remained <15%; the recovery strategy was considered acceptable if it restored steady-state exposure without exceeding a mean $AUC_{0-12h}$ increase of 30%.

## 3 Results

### 3.1 Exposure at steady state

At steady state, the mean $AUC_{0-12h}$ increased proportionally with MMF dose (Fig 2A-B; Table 2). Using Rong's model, the mean $AUC_{0-12h}$ was 31.8 ± 7.6, 47.6 ± 10.3, and 63.5 ± 15.3 mg·h/L for 500, 750, and 1000 mg BID, respectively. With van Hest's model, corresponding values were 35.5 ± 17.3, 47.7 ± 22.8, and 59.1 ± 28.7 mg·h/L for 750, 1000, and 1250 mg BID.

Despite therapeutic dosing, the interindividual variability was substantial and underexposure was frequent at steady state, particularly with lower doses: 42.7% of the $AUC_{0-12h}$ values were <30 mg·h/L for 500 mg BID (Rong) and 42.1% for 750 mg BID (van Hest).

### 3.2 Impact of a fully missed dose

Missed doses resulted in substantial exposure drops. A single missed MMF dose resulted in a marked decrease in MPA exposure across all dose groups and both models (Fig 3A–B, Table 2). The mean $AUC_{0-12h}$ reductions ranged from 40% to over 60%, with the most pronounced drop observed in the lower dose groups. For instance, following a missed 750 mg

**Table 1. Summary of MMF pharmacokinetic models used for simulation[a].**

|  | Rong et al. [27] | Van Hest et al. [28] |
|---|---|---|
| Structural model | Two-compartment model, first-order absorption with a lag time, linear elimination | Two-compartment model with time lagged first order absorption, linear elimination |
| Software | MonolixSuite-2018R1 | NONMEM |
| Pharmacokinetic parameters | $Ka = 1.98\,h^{-1}$<br>$V_1/F = 25\,L$<br>$V_2/F = 607\,L$<br>$CL/F = 2.87\,L/h$<br>$Q/F = 36.7\,L/h$<br>$Tlag = 0.162\,h$ | $Ka = 4.1\,h^{-1}$<br>$V_1/F = 91\,L$<br>$V_2/F = 237\,L$<br>$CL/F = 33\,L/h$<br>$Q/F = 35\,L/h$<br>$Tlag = 0.21\,h$ |
| Model variability | $\omega\,Ka = 0.99$<br>$\omega\,V_1/F = 0.18$<br>$\omega\,V_2/F = 1.08$<br>$\omega\,CL/F = 0.23$<br>$\omega\,Q/F = 0.27$<br>$\omega\,Tlag = 1.08$<br>Original proportional error $\sigma = 0.32$<br>Original additive error $\sigma = 0.08\,mg/L$<br>Decreased residual error $\sigma$: 0.001 | $\omega\,Ka = 0.89$<br>$\omega\,V_1/F = 0.77$<br>$\omega\,V_2/F = 0.84$<br>$\omega\,CL/F = 0.30$<br>$\varkappa\,Ka = 0.93$<br>$\varkappa\,V_1/F = 0.62$<br>$\varkappa\,CL/F = 0.29$<br>Original additive error $\sigma = 0.45\,mg/L$<br>Decreased residual error $\sigma$: 0.001 |
| Covariates | Covariates on CL/F<br><br>• AcMPAG concentration<br>• MPAG/MPA AUC ratio | Covariates on $V_1/F$<br><br>• ClCr<br>• Alb<br><br>Covariates on CL/F<br><br>• Gender<br>• $CL_{Cr}$<br>• Albumin<br>• Cicosplorin dose |

[a]Abbreviations: Ka: absorption rate constant; V1/F: apparent central volume of distribution; V2/F: apparent peripheral compartment volume of distribution; CL/F: apparent central clearance; Q/F: apparent inter-compartment clearance; Tlag: absorption lag time; $\varkappa$: intra-patient variability; $\omega$: inter-individual variability (SD); $\sigma$: residual error.

AcMPAG: mycophenolic acid acyl-glucuronide; MPAG: mycophenolic acid glucuronide; MPA: mycophenloc acid; ClCr: creatinine clearance; Alb: plasma albumin concentration.

dose, the mean $AUC_{0-12h}$ decreased from $47.6 \pm 10.3$ mg·h/L to $28.0 \pm 11.2$ mg·h/L (Rong) and from $35.5 \pm 17.3$ mg·h/L to $17.9 \pm 14.3$ mg·h/L (van Hest), corresponding to relative reductions of 41.2% and 49.6%, respectively. The proportion of patients with an $AUC_{0-12h} < 30$ mg·h/L rose to 60.4% with Rong's model and 83.2% with van Hest's model. Similar trends were observed in the other dose groups.

Following a missed dose, the time to return to steady-state exposure was also prolonged. For the 750 mg BID dose, the median recovery time was 48 hours for both models, with a maximum of 72 hours (Rong) and 84 hours (van Hest).

### 3.3 Impact of delayed dose intake

Delaying a dose had a graded impact on MPA exposure (Table 2, Fig 3C–D). Delays up to 6 hours resulted in <15% reduction in the mean $AUC_{0-12h}$, with a return to steady state generally achieved within 24–36 hours.

In low-dose groups, a 6-hour delay reduced the mean $AUC_{0-12h}$ to $28.8 \pm 7.2$ mg·h/L (Rong, 500 mg BID) and $30.4 \pm 14.7$ mg·h/L (van Hest, 750 mg BID), with underexposure observed in over 55% of patients. Longer delays (>6 hours) caused more pronounced drops in exposure (up to approx. 35%), and delayed recovery to steady state (≥48 h).

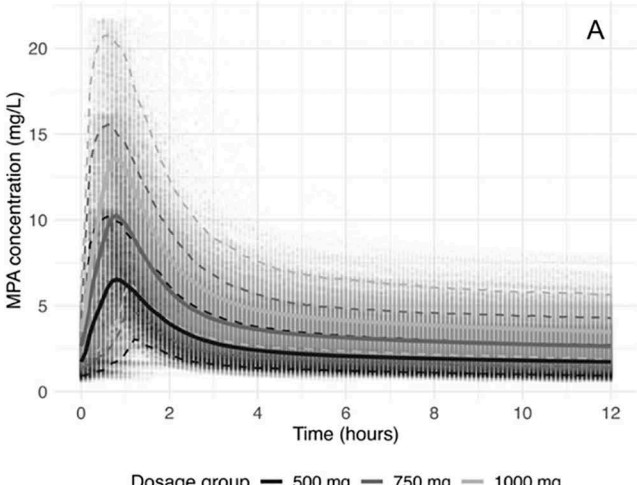
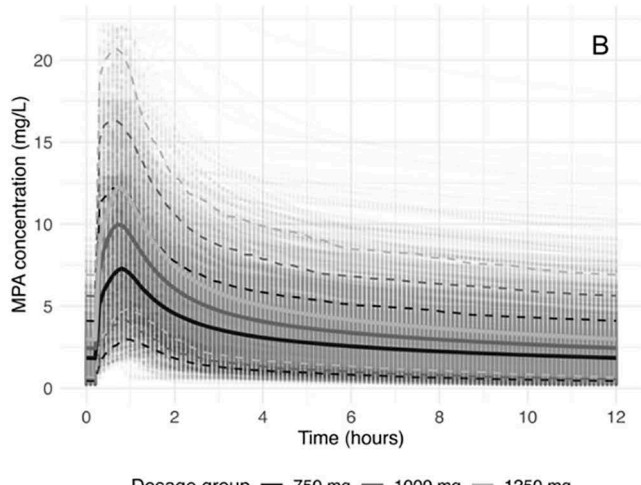

**Fig 2. Simulated pharmacokinetic profiles of MPA.** Simulated steady-state pharmacokinetic profiles of MPA (mean -solid lines- and 5th-95th percentiles -dashed lines) for three MMF BID dosing regimens using Rong's model (left (A)) and van Hest's model (right (B)).

### 3.4 Recovery strategies after a missed dose

Recovery strategies consisting of either adding 50% or 100% of the missed dose at the next scheduled intake were tested in each group (Table 3, Fig 3E–H).

Simulated $AUC_{0-12h}$ values following recovery strategies (50% or 100% compensation at next intake). Includes proportions of patients outside the therapeutic window and relative changes from baseline exposure (%). NA = not applicable; strategies are rated based on AUC rebound and proportion of overexposure.

Full-dose compensation restored the $AUC_{0-12h}$ but often overshot the therapeutic range. The mean $AUC_{0-12h}$ increased by 30–40%, with up to 80% of patients exceeding 60 mg.h/L, particularly in higher-dose groups. This strategy allowed a return to steady state within a median time of 48 hours.

Half-dose compensation offered a more balanced exposure profile: this approach increased the $AUC_{0-12h}$ by 8–13% on average and limited the proportion of patients above the upper target to <20%, with recovery to steady state achieved in 36–48 hours.

Dose rounding strategies were tested in dosing regimens without commercially available 50% increments (e.g., 750 mg or 1250 mg BID). For 750 mg BID, adding 250 mg (rounded down) led to modest AUC increases (+2.5 to +3.5%) and acceptable target range maintenance. In contrast, adding 500 mg (rounded up) led to larger increases (up to +22%) and a higher risk of overexposure. Similarly, in the 1250 mg BID group, rounding up to a 750 mg added dose led to >55% of patients exceeding 60 mg·h/L, whereas a 500 mg add-on limited this risk to 46.2%. In both situations, the median time to recover steady state was 36–48 hours. Overall, rounding down when half-doses are unavailable provided a better balance between appropriate exposure and overexposure.

## 4 Discussion

This simulation-based study provides a quantitative assessment of the pharmacokinetic consequences of missing or delaying doses of mycophenolate mofetil (MMF) in kidney transplant recipients and evaluates practical strategies to mitigate these effects. Although non-adherence is a well-recognized contributor to poor transplant outcomes, few studies have explored, in a model-informed manner, how specific deviations from immunosuppressive regimens impact drug

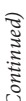

Table 2. MPA AUC$_{0-12h}$ under different dosing scenarios.

| Dosage (mg BID) | Time between the last dose at steady state and the following dose intake | Rong et al. [27] AUC$_{0-12h}$ (mg.h/L) | Proportion of patients with AUC$_{0-12h}$ (%) <30 mg.h/L | 30-60 mg.h/L | >60 mg.h/L | Relative difference vs. the reference AUC$_{0-12h}$ (%) | van Hest et al. [28] AUC$_{0-12h}$ (mg.h/L) | Proportion of patient with AUC$_{0-12h}$ (%) <30 mg.h/L | 30-60 mg.h/L | >60 mg.h/L | Relative difference vs. the reference AUC$_{0-12h}$ (%) | Evaluation of strategy |
|---|---|---|---|---|---|---|---|---|---|---|---|---|
| **Steady state (reference)** | | | | | | | | | | | | |
| 500 | 12 | 31.8±7.6 | 42.7 | 57.3 | 0.0 | NA | NA | | | | NA | NA |
| 750 | 12 | 47.6±10.3 | 3.0 | 83.4 | 13.6 | | 35.5±17.3 | 42.1 | 49.7 | 8.2 | NA | NA |
| 1000 | 12 | 63.5±15.3 | 0.2 | 46.8 | 53.0 | | 47.7±22.8 | 19.9 | 57.0 | 23.1 | | |
| 1250 | 12 | NA | | | | | 59.1±28.7 | 10.6 | 49.4 | 40.0 | | |
| **Fully missed dose** | | | | | | | | | | | | |
| 500 | 24 | 18.1±7.5 | 92.8 | 7.2 | 0.0 | -43.1±15.4 | NA | | | | NA | |
| 750 | 24 | 28.0±11.2 | 60.4 | 38.5 | 1.1 | -42.1±14.8 | 17.9±14.3 | 83.2 | 15.3 | 0.5 | -56.5±18.2 | |
| 1000 | 24 | 36.2±15.2 | 33.3 | 60.2 | 6.5 | -43.3±15.7 | 23.6±18.3 | 71.2 | 23.7 | 5.1 | -56.4±17.9 | |
| 1250 | 24 | NA | | | | | 28.1±23.5 | 63.4 | 28.9 | 7.7 | -57.7±18.2 | |
| **Delayed dose** | | | | | | | | | | | | |
| 500 | 14 | 31.0±7.6 | 46.2 | 53.8 | 0.0 | -2.6±2.2 | NA | | | | | Acceptable exposure |
| | 16 | 30.0±7.5 | 51.7 | 48.3 | 0.0 | -5.4±7.5 | | | | | | Acceptable exposure |
| | 18 | 28.8±7.2 | 55.9 | 43.9 | 0.02 | -8.8±6.5 | | | | | | Acceptable exposure |
| | 20 | 27.6±7.6 | 64.2 | 35.7 | 0.01 | -13.0±9.3 | | | | | | Underexposure |
| | 22 | 24.6±7.8 | 75.7 | 24.3 | 0.0 | -21.8±13.1 | | | | | | Underexposure |
| 750 | 14 | 46.4±11.3 | 4.1 | 84.9 | 11 | -2.6±2.0 | 34.0±16.7 | 46.9 | 46.6 | 6.5 | -4.1±2.3 | Acceptable exposure |
| | 16 | 44.9±11.2 | 7.1 | 83.5 | 9.4 | -5.4±3.9 | 32.5±16.0 | 50.2 | 44.4 | 5.4 | -9.0±3.6 | Acceptable exposure |
| | 18 | 43.2±10.8 | 8.8 | 85.4 | 5.8 | -8.8±6.4 | 30.4±14.7 | 56.5 | 39.1 | 4.4 | -14.7±5.7 | Acceptable exposure |
| | 20 | 41.3±11.3 | 14.0 | 79.9 | 6.1 | -13.1±9.0 | 27.8±14.4 | 64.8 | 31.8 | 3.4 | -22.0±8.4 | Underexposure |
| | 22 | 36.8±11.7 | 27.4 | 69.1 | 3.5 | -21.9±13.1 | 24.1±14.7 | 73.2 | 24.1 | 2.7 | -35.3±13.2 | Underexposure |

(Continued)

**Table 2.** (Continued)

| | | Rong et al. [27] | | | | | van Hest et al. [28] | | | | | |
| Dosage (mg BID) | Time between the last dose intake at steady state and the following dose intake | AUC$_{0-12h}$ (mg.h/L) | Proportion of patients with AUC$_{0-12h}$ (%) | | | Relative difference vs. the reference AUC$_{0-12h}$ (%) | AUC$_{0-12h}$ (mg.h/L) | Proportion of patient with AUC$_{0-12h}$ (%) | | | Relative difference vs. the reference AUC$_{0-12h}$ (%) | Evaluation of strategy |
| | | | < 30 mg.h/L | 30-60 mg.h/L | > 60 mg.h/L | | | < 30 mg. h/L | 30-60 mg. h/L | > 60 mg. h/L | | |
| 1000 | 14 | 61.9±15.0 | 0.2 | 50.4 | 49.4 | -2.6±1.9 | 45.8±22.1 | 19.2 | 55.7 | 25.1 | -4.0±1.8 | Acceptable exposure |
| | 16 | 59.9±14.9 | 0.4 | 55.7 | 43.9 | -5.5±3.9 | 43.2±20.0 | 23.4 | 60.4 | 16.2 | -8.7±3.4 | Acceptable exposure |
| | 18 | 57.6±14.4 | 0.8 | 59.4 | 39.8 | -8.8±6.3 | 40.1±19.5 | 33.0 | 52.0 | 15.0 | -14.5±5.6 | Acceptable exposure |
| | 20 | 55.1±15.1 | 2.3 | 64.9 | 32.8 | -13.1±9.4 | 37.4±18.9 | 39.3 | 50.8 | 9.9 | -22.1±8.5 | Underexposure |
| | 22 | 49.1±15.6 | 9.2 | 69.5 | 21.3 | -21.8±13.1 | 30.4±18.0 | 57.3 | 37.1 | 5.6 | -35.8±13.3 | Underexposure |
| 1250 | 14 | NA | | | | | 56.7±27.8 | 12.8 | 51.0 | 36.2 | 4.1±1.7 | Acceptable exposure |
| | 16 | | | | | | 54.2±26.6 | 14.5 | 53.0 | 32.5 | -9.0±3.4 | Acceptable exposure |
| | 18 | | | | | | 50.6±24.5 | 18.4 | 53.5 | 28.1 | -14.8±5.5 | Acceptable exposure |
| | 20 | | | | | | 46.3±24.1 | 24.3 | 53.7 | 22.0 | -22.1±8.4 | Underexposure |
| | 22 | | | | | | 40.1±24.5 | 40.6 | 43.0 | 16.4 | -35.3 ± 12.6 | Underexposure |

Mean AUC$_{0-12h}$ (± SD), proportion of patients (%) below therapeutic threshold (< 30 mg.h/L), and relative difference from steady-state exposure (%) are reported for each dose and model. Results are grouped by scenario: steady state, missed dose, and delayed dose. Highlighted cells may indicate unacceptable exposure levels.

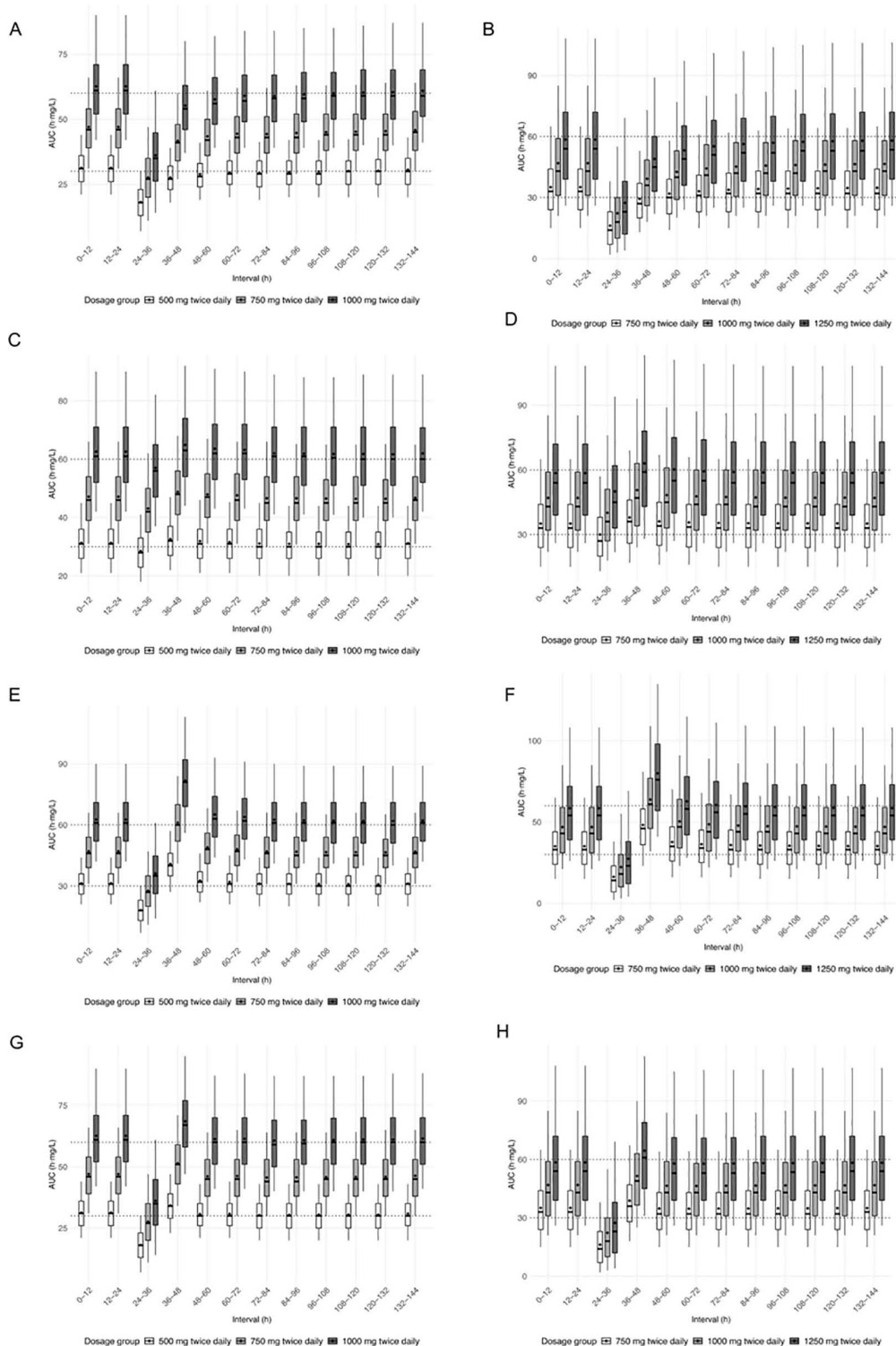

**Fig 3. MPA AUC$_{0-12h}$ over time following missed or 6-hours delayed MMF doses.** Panels display AUC$_{0-12h}$ box-and-whisker plots over time to assess recovery following different scenarios. Time 0 corresponds to the steady state. **Panels A-C-E-G**: simulations based on the **Rong's model**. **Panels B-D-F-H**: simulations based on the **Van Hest's model**. Scenarios illustrated: (A-B) Missed dose at 24 h with no compensation strategy. (C-D) 6-hour delayed dose (intake at 30 h). (E-F) Full missed dose added at next intake (200% dose at 36 h). (G-H) Half missed dose added at next intake (150% dose at 36 h).

**Table 3. Effectiveness of recovery strategies following missed dose.**

| Dosage (mg BID) | Recovery dose* (mg) | Rong et al. (27) AUC$_{0-12h}$ (mg.h/L) | Proportion of patients with AUC$_{0-12h}$ (%) | | | Relative difference vs. the reference AUC$_{0-12h}$ (%) | van Hest et al. (28) AUC$_{0-12h}$ (mg.h/L) | Proportion of patient with AUC$_{0-12h}$ (%) | | | Relative difference vs. the reference AUC$_{0-12h}$ (%) | Evaluation of strategy Relative difference vs. the reference AUC$_{0-12h}$ (%) |
|---|---|---|---|---|---|---|---|---|---|---|---|---|
| **Fully missed dose compensated by the administration of a supplementary half dose at the next scheduled intake (administration of a 150% dose)** | | | | | | | | | | | | |
| 500 | 500+250 | 34.5±8.2 | 27.8 | 71.9 | 0.03 | +8.5±6.6 | NA | | | | | Acceptable exposure |
| 750 | *150% dose impossible* | NA | | | | | NA | | | | | |
| | 750+250 | 48.6±11.7 | 3.4 | 81.5 | 15.4 | +2.5±5.6 | 36.1±16.1 | 38.2 | 55.8 | 6.0 | +3.5±8.6 | Acceptable exposure |
| | 750+500 | 54.2±12.4 | 0.8 | 70.5 | 28.7 | +16.9±11.4 | 42.3±16.8 | 21.9 | 64.6 | 13.5 | +22.3±14.0 | Overexposure |
| 1000 | 1000+500 | 68.7±16.2 | 0.1 | 32.2 | 67.7 | +8.1±6.7 | 53.4±22.1 | 9.9 | 58.7 | 31.4 | +12.5±10.9 | Acceptable exposure |
| 1250 | *150% dose impossible* | NA | | | | | NA | | | | | |
| | 1250+500 | NA | | | | | 62.2±27.1 | 6.3 | 47.5 | 46.2 | +6.5±8.6 | Acceptable exposure |
| | 1250+750 | | | | | | 68.4±27.3 | 3.9 | 40.3 | 55.8 | +19.1±12.0 | Overexposure |
| **Fully missed dose followed by the administration of a supplementary full dose at the next scheduled intake (administration of a 200% dose)** | | | | | | | | | | | | |
| 500 | 500+500 | 40.6±9.3 | 10.4 | 87.2 | 2.4 | +31.1±13.1 | NA | | | | | Overexposure |
| 750 | 750+750 | 61.4±14.4 | 0.05 | 50.6 | 48.9 | +30.7±12.9 | 48.5±19.3 | 13.6 | 64.4 | 22.0 | +41.2±19.6 | Overexposure |
| 1000 | 1000+1000 | 81.5±18.8 | 0.1 | 11.8 | 88.1 | +31.3±13.3 | 66.5±25.9 | 3.4 | 42.3 | 54.3 | +40.1±18.8 | Overexposure |
| 1250 | 1250+1250 | NA | | | | | 80.7±32.4 | 1.6 | 26.9 | 80.7 | +41.5±19.5 | Overexposure |

*The recovery dose corresponds to the total dose administered after a missed dose.

exposure and how best to respond to them. Prior research has addressed recovery strategies for tacrolimus [30], but this is, to our knowledge, the first study to explore these aspects for MMF using model-based simulations.

Our results demonstrate that a single missed dose of MMF has a substantial and prolonged effect on mycophenolic acid (MPA) exposure. Regardless of the dose or pharmacokinetic model applied, the mean AUC$_{0-12h}$ dropped by 40–60%, and the majority of patients fell below the recommended therapeutic range of 30–60 mg·h/L(1). Notably, recovery to baseline exposure was not immediate: it required on average two full dosing intervals, with some individuals needing up to 72–84 hours to return to steady-state. This suggests that a missed dose should not be considered a transient event without consequence, especially in patients with low baseline exposure or high immunologic risk.

These findings are clinically relevant, particularly given the known association between low MPA exposure and increased risk of acute rejection, especially in the early post-transplant period [16]. They also align with earlier studies on tacrolimus showing that dose omissions, even when infrequent, can induce significant fluctuations in drug levels [31,32], with potential clinical repercussions. However, unlike tacrolimus, MMF has a longer half-life and is often administered at fixed doses without therapeutic drug monitoring (TDM) in routine practice. Our results support a more individualized approach to MMF dosing, including targeted patient education and clear recovery instructions in case of omissions.

In contrast, delays of up to 6 hours had a limited effect on exposure (<15% decrease in AUC), with rapid return to steady state (≤ 36 hours). These findings support current recommendations that MMF doses may be taken later the same day if the delay does not exceed 6 hours. Conversely, longer delays behave more like full omissions from a pharmacokinetic standpoint and may warrant compensatory strategies.

Our simulations show that full-dose compensation (*i.e.*, doubling the next intake) is not advisable, as it leads to a sharp rebound in exposure, with frequent overshooting beyond the therapeutic window, especially in patients receiving higher MMF doses. This supports current label recommendations discouraging double dosing. This risk is further amplified in patients exhibiting high interindividual variability, in whom drug accumulation and unpredictable exposure peaks may occur. In contrast, a more conservative approach -adding half of the missed dose at the next scheduled intake- offered a more favorable balance: it restored $AUC_{0-12h}$ to within target limits in over 95% of cases, across all dosing regimen and PK models without overshooting, thus minimizing the risk of toxicity. This strategy may therefore serve as a pragmatic, standardized recommendation for clinical practice. A summary table of these model-informed recommendations is provided below (Table 4). Integrating these recommendations into patient education resources, discharge procedures, or therapeutic drug monitoring practices could enhance their clinical applicability and support standardized responses to missed or delayed doses in routine care.

Importantly, it balances the risk of underexposure and its potential consequences for graft rejection against the minimal risk associated with modest compensatory dosing.

This study also addresses a gap frequently raised by clinical teams, who often lack clear, evidence-based guidance for MMF dose omissions. In the absence of published data, our model-informed approach offers a pragmatic, practical tool to support adherence counseling and therapeutic decision-making in routine care. Based on available safety data, we estimate that the risk of gastrointestinal adverse events associated with a temporary 50% dose increase remains very low, particularly when used occasionally as a recovery measure [33].

We also addressed a common clinical issue: the lack of intermediate-dose formulations. In clinical practice, MMF is available in 250 mg and 500 mg capsules or tablets, which limits flexibility when attempting to compensate precisely for a missed dose. For 750 mg and 1250 mg BID regimens, we tested dose rounding strategies. In both cases, administering the lower available dose (250 mg instead of 375 mg and 500 mg instead of 625 mg, respectively) resulted in a minimal under-correction while avoiding the risks associated with overexposure. Rounding up, in contrast, increased the likelihood of exceeding the upper therapeutic threshold, especially in high-dose regimens. These results provide practical, model-informed recommendations for clinical management.

Beyond the evaluation of missed doses, our study also confirms a well-documented but often underestimated phenomenon: the high interindividual variability of MPA exposure, even at steady state. This variability was especially pronounced in the van Hest model, which reflects patients co-treated with ciclosporin. Inhibitory effects of ciclosporin on enterohepatic recirculation of MPA likely explain this difference, and the clinical consequence is a greater risk of underexposure in this population, particularly when a dose is missed [34,35]. These results further advocate for routine TDM in patients receiving MMF, especially in the context of suspected non-adherence or concomitant drug interactions [36].

**Table 4. Practical, model-informed recommendations for managing missed or delayed MMF doses.**

| Clinical scenario | Recommended Action | Rationale |
|---|---|---|
| Delay<6 hours | Take the full dose as soon as the delay is identified | Has a minimal impact on the AUC; supports existing clinical recommendations |
| Delay>6 hours or Fully missed dose | At the time of the intake following the delayed or missed dose: add half of the missed dose to the regular dose | Restores $AUC_{0-12h}$ and limits the risk of underexposure |
| No half dose formulation available | Round down to the nearest available dose (e.g., 250 mg) | Reduces the risk of overexposure |

The study has some limitations that should be acknowledged. Firstly, it relied on two population PK models derived from specific cohorts of adult kidney transplant patients (treated with either tacrolimus or ciclosporin). Extrapolation to other populations (e.g., liver transplant, pediatric patients, or recipients on alternative co-medications such as rifampicin) should be done with caution. In such cases, therapeutic drug monitoring remains essential to ensure adequate exposure. Secondly, in our simulations, we fixed the residual unexplained variability to a minimal value ($\sigma = 0.001$) to specifically focus on the consequences of inter-individual pharmacokinetic variability and adherence patterns. This may have led to an underestimation of the overall variability observed in clinical practice, especially in patients with fluctuating adherence or altered absorption. Thirdly, to reduce the influence of extreme simulations with unrealistic parameter combinations, AUC values below the 0.1st and above the 99.9th percentiles were excluded, following general methodological recommendations for outlier handling. Nonetheless, we acknowledge that this may slightly limit the generalizability of our results to patients at the extremes of pharmacokinetic variability. It is important to note, however, that such extreme profiles were rare in our simulations and represent only a very small fraction of the overall population (<0.2%). Importantly, individuals with low baseline exposure who tend to fall in the lower tail of the AUC distribution are also those most impacted by missed doses, with slower recovery and a greater risk of underexposure. These patients remain a key target population for therapeutic drug monitoring and enhanced adherence support. Conversely, individuals with very high baseline exposure (upper tail) are less vulnerable to the impact of a missed dose and are at lower risk of falling below the therapeutic window. Notably, our proposed compensatory strategy adding half the missed dose did not lead to simulated exposures exceeding toxic thresholds, even in high-exposure profiles. Therefore, while extreme profiles were not formally retained in the analysis, the simulations still capture the clinically relevant range of variability and support the robustness and safety of our recommendations across most patient scenarios.

We also acknowledge the lack of external validation of our simulations; however, prospective validation in clinical settings would raise ethical concerns, particularly by deliberately inducing missed doses in transplant recipients. Finally, our study focused on isolated dosing deviations (i.e., single missed or delayed doses). In clinical practice, however, non-adherence is often recurrent, intermittent, or fluctuating over time particularly among certain subgroups such as adolescents, patients with limited health literacy, or those facing complex treatment regimens. Simulating such dynamic and individualized adherence patterns poses methodological challenges, especially in terms of variability propagation and long-term pharmacokinetic impact. Therefore, evaluating the cumulative consequences of repeated or clustered deviations, and the potential role of adaptive or personalized recovery strategies, could further enhance the clinical relevance and applicability of model-informed approaches in transplant medicine.

Despite these limitations, our study is, to our knowledge, the first to offer a comprehensive, simulation-based evaluation of MMF dose omissions and proposes pharmacologically sound, clinically actionable recovery strategies. It provides a framework that could be integrated into patient education tools, mobile health applications, or clinical decision support systems. The adoption of such strategies by clinical teams may help fill a critical gap in current practice, especially in the absence of formal guidelines for the management of missed MMF doses. Future directions include the validation of these model-informed recommendations in clinical settings, ideally through prospective trials or real-world observational cohorts. Extending this approach to other immunosuppressive drugs and transplant populations could also help develop harmonized, personalized adherence management strategies.

## 5 Conclusion

In the context of long-term immunosuppressive therapy, missed doses are a foreseeable reality-even among well-informed and adherent transplant recipients. This study demonstrates that omitting a single dose of MMF can result in significant and prolonged underexposure to MPA, particularly in patients receiving lower MMF doses or co-treated with ciclosporin. Such underexposure may persist for up to three days, potentially compromising immunosuppressive efficacy and increasing the risk of rejection.

Using two validated population pharmacokinetic models, we simulated a range of clinical scenarios and identified pragmatic, model-informed strategies to mitigate the impact of MMF dose omissions. In cases of short delays (≤6 hours), taking the delayed dose without adjustment was found to be pharmacokinetically acceptable. When a full dose is missed, compensating with half of the missed dose at the next scheduled intake effectively restored exposure while avoiding over-correction. When exact dose adjustments are not feasible due to formulation constraints, rounding down to the nearest lower available dose appears safer than rounding up.

These findings support the development of clear, evidence-based recommendations to guide clinical decision-making and patient counseling in the event of MMF dose deviations. They also underscore the value of individualized therapeutic drug monitoring in transplant recipients, particularly those at risk of non-adherence or altered pharmacokinetics.

Future studies are warranted to validate these recommendations prospectively and assess their applicability in other transplant populations. Ultimately, the integration of such model-informed strategies into clinical guidelines and digital adherence tools could enhance the safety and precision of immunosuppressive therapy.

## Supporting information

**S1 File.  Raw data allowing replicating the results of this study.**
(PDF)

## Acknowledgments

The authors thank the simulation and modeling support team at CHU Limoges for their technical assistance in implementing the population pharmacokinetic models.

## Author contributions

**Conceptualization:** Caroline Monchaud.

**Data curation:** Franck Maizaud, Selim Arraki-Zava, Hamza Sayadi, Yeleen Fromage.

**Formal analysis:** Franck Maizaud.

**Methodology:** Selim Arraki-Zava, Hamza Sayadi, Yeleen Fromage, Pierre Marquet, Jean-Baptiste Woillard.

**Project administration:** Caroline Monchaud.

**Resources:** Pierre Marquet.

**Software:** Selim Arraki-Zava, Hamza Sayadi, Yeleen Fromage, Jean-Baptiste Woillard.

**Supervision:** Pierre Marquet, Jean-Baptiste Woillard, Caroline Monchaud.

**Visualization:** Franck Maizaud.

**Writing – original draft:** Franck Maizaud.

**Writing – review & editing:** Jean-Baptiste Woillard, Caroline Monchaud.

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
