## [Decision Letter · Decision Letter 0]

22 Jul 2025

Dear Dr. Monchaud,

Thank you for submitting your manuscript to PLOS ONE. After careful consideration, we feel that it has merit but does not fully meet PLOS ONE’s publication criteria as it currently stands. Therefore, we invite you to submit a revised version of the manuscript that addresses the points raised during the review process.

The authors are kindly requested to address the reviewers' comments - particularly those from Reviewer 2 - in a timely and thorough manner before proceeding with the next stage of the PLOS ONE review process.

We look forward to receiving your revised manuscript.

Kind regards,

Paulo Caceres Guido

Academic Editor

PLOS ONE

Journal Requirements:

“I have read the journal's policy and the authors of this manuscript have the following competing interests:

FM, SAZ, HS, and YF declare no competing interests.

CM has received speaker honoraria and/or research funding from Chiesi, Astellas, Pfizer, and Gilead.

PM has received speaker and consulting honoraria and/or research grants from MedinCell, Chiesi, Sandoz, Astellas, and Bristol Myers Squibb (BMS).

JBW has received speaker honoraria from Chiesi, Astellas, Pfizer, Vifor, and Gilead”

5. We note you have included a table to which you do not refer in the text of your manuscript. Please ensure that you refer to Table 1, 2 and 3 in your text; if accepted, production will need this reference to link the reader to the Table.

Reviewers' comments:

Reviewer's Responses to Questions

**Comments to the Author**

1. Is the manuscript technically sound, and do the data support the conclusions?

Reviewer #1: Yes

Reviewer #2: Yes

2. Has the statistical analysis been performed appropriately and rigorously?

Reviewer #1: Yes

Reviewer #2: Yes

3. Have the authors made all data underlying the findings in their manuscript fully available?

Reviewer #1: Yes

Reviewer #2: Yes

4. Is the manuscript presented in an intelligible fashion and written in standard English?

Reviewer #1: Yes

Reviewer #2: Yes

Reviewer #1: Dear Author/s

Greetings

The incidence of drug-related rejection (overdose or missed medication) is significant in renal transplantation.

Calcineurin levels can be monitored at 0 and 2 hours during medication monitoring. However, mycophenolate mofetil levels did not check and are ignore.

The study also carefully monitored and analyzed the AUC levels and efficacy of missed medication.

The article is suitable for publication in its current form.

Reviewer #2: This manuscript investigates the pharmacokinetic consequences of missed or delayed doses of mycophenolate mofetil (MMF) in kidney transplant recipients, using simulations based on two previously published and validated population pharmacokinetic (PopPK) models. The authors explore a range of clinically relevant scenarios — including full dose omissions, delays of varying duration, and compensation strategies — across multiple dosing regimens. The study shows that a fully missed dose leads to a 40–60% reduction in MPA exposure (AUC₀–₁₂h) and a delayed return to steady state (up to 72–84 hours). Delays of less than 6 hours had only a limited impact on exposure (<15% reduction), while full-dose compensation frequently resulted in overexposure, particularly in higher-dose regimens. In contrast, a strategy of administering 50% of the missed dose at the next intake effectively restored therapeutic exposure without excessive rebound.

The topic is highly relevant for clinicians involved in transplantation. Missed or delayed immunosuppressive doses are common in clinical practice, yet current guidance remains vague and largely unsupported by pharmacokinetic evidence. This study fills an important gap by providing quantitative, model-based recommendations that can guide therapeutic decision-making in routine care. The modeling approach is methodologically appropriate, well described, and transparently reported. The clinical scenarios are realistic and aligned with the questions encountered by physicians and pharmacists in the field.

Some clarifications or additions, however, would improve the manuscript's clinical applicability and methodological transparency :

1/ Although the results are clearly presented, their practical implementation could be further supported. The key recommendations would benefit from being summarized in an accessible format, such as a decision table or clinical algorithm. A brief discussion of how these strategies could be integrated into patient education materials, discharge protocols, or therapeutic drug monitoring procedures would help reinforce their clinical utility.

2/ The authors fixed the residual error at a minimal value (σ = 0.001) to focus on inter-individual variability. While this is justifiable in the context of simulation, it likely underestimates the overall variability encountered in clinical settings, particularly among patients with fluctuating adherence or altered absorption. This limitation should be acknowledged explicitly, as it may affect the generalizability of the findings.

3/ AUC values below the 0.1st and above the 99.9th percentiles were excluded, with reference to general recommendations on outlier handling. Although this is methodologically reasonable, the implications of this decision should be briefly discussed. Patients with extreme pharmacokinetic profiles — such as those with low baseline exposure — may be at greater risk of adverse consequences following a missed dose. Excluding these individuals could potentially limit the applicability of the findings to high-risk subgroups.

4/ The study models isolated dosing deviations. In clinical practice, however, non-adherence is frequently recurrent or fluctuating. While such patterns are more complex to simulate, this should be noted as a limitation, and future work exploring the impact of repeated deviations could be proposed.

**Do you want your identity to be public for this peer review?** For information about this choice, including consent withdrawal, please see our Privacy Policy

Reviewer #1: **Yes: ** Yavuz Ayar

Reviewer #2: No

---

## [Author Response · Author response to Decision Letter 1]

4 Aug 2025

We thank the reviewers and the academic editor for their careful evaluation of our manuscript. Below, we provide detailed responses to each of the comments raised. The manuscript has been revised in accordance with the comments made.

Editor

=> Done.

“I have read the journal's policy and the authors of this manuscript have the following competing interests:

FM, SAZ, HS, and YF declare no competing interests.

CM has received speaker honoraria and/or research funding from Chiesi, Astellas, Pfizer, and Gilead.

PM has received speaker and consulting honoraria and/or research grants from MedinCell, Chiesi, Sandoz, Astellas, and Bristol Myers Squibb (BMS).

JBW has received speaker honoraria from Chiesi, Astellas, Pfizer, Vifor, and Gilead”

Please confirm that this does not alter your adherence to all PLOS ONE policies on sharing data and materials, by including the following statement: "This does not alter our adherence to PLOS ONE policies on sharing data and materials.” If there are restrictions on sharing of data and/or materials, please state these. Please note that we cannot proceed with consideration of your article until this information has been declared.

=> Done (no restrictions on sharing of data and/or materials).

Please confirm at this time whether or not your submission contains all raw data required to replicate the results of your study.

=> All raw data required to replicate the results of your study are within the manuscript and its Supporting Information files; this statement has been inserted in the cover letter.

4. PLOS requires an ORCID iD for the corresponding author in Editorial Manager on papers submitted after December 6th, 2016. Please ensure that you have an ORCID iD and that it is validated in Editorial Manager.

=> Done.

5. We note you have included a table to which you do not refer in the text of your manuscript. Please ensure that you refer to Table 1, 2 and 3 in your text; if accepted, production will need this reference to link the reader to the Table

=> All the tables are referred to in the manuscript.

=> Not applicable

7. Please review your reference list to ensure that it is complete and correct.

=> Done.

Reviewer #1

This reviewer supported the publication of the manuscript in its current form. We thank them sincerely for their positive feedback and appreciation of the clinical relevance of our study.

Reviewer #2

1/ Although the results are clearly presented, their practical implementation could be further supported. The key recommendations would benefit from being summarized in an accessible format, such as a decision table or clinical algorithm. A brief discussion of how these strategies could be integrated into patient education materials, discharge protocols, or therapeutic drug monitoring procedures would help reinforce their clinical utility.

Response:

We agree with the reviewer that the clinical implementation of our findings is essential. We have added a clear reference to a new decision table (Table 4) summarizing the key model-informed recommendations. We also added a sentence explicitly discussing how these strategies could be integrated into clinical practice.

2/ The authors fixed the residual error at a minimal value (σ = 0.001) to focus on inter-individual variability. While this is justifiable in the context of simulation, it likely underestimates the overall variability encountered in clinical settings, particularly among patients with fluctuating adherence or altered absorption. This limitation should be acknowledged explicitly, as it may affect the generalizability of the findings.

Response:

We appreciate this important point and have clarified this limitation in the revised manuscript. We now explicitly state that fixing the residual error may underestimate real-life variability, especially in patients with altered adherence or absorption.

3/ AUC values below the 0.1st and above the 99.9th percentiles were excluded, with reference to general recommendations on outlier handling. Although this is methodologically reasonable, the implications of this decision should be briefly discussed. Patients with extreme pharmacokinetic profiles such as those with low baseline exposure may be at greater risk of adverse consequences following a missed dose. Excluding these individuals could potentially limit the applicability of the findings to high-risk subgroups.

Response:

We thank the reviewer for highlighting this. We have now included a discussion of the implications of excluding extreme AUC values, emphasizing that such profiles were rare (<0.2%), and that patients with low exposure remain a key target for TDM. We also highlight that the compensatory strategy was safe even in high-exposure profiles.

4/ The study models isolated dosing deviations. In clinical practice, however, non-adherence is frequently recurrent or fluctuating. While such patterns are more complex to simulate, this should be noted as a limitation, and future work exploring the impact of repeated deviations could be proposed.

Response:

We agree with the reviewer that recurrent or fluctuating non-adherence is highly relevant in real-world settings. We now explicitly acknowledge this limitation and propose it as an important direction for future research.

---

## [Editor Report · Decision Letter 1]

7 Aug 2025

Population Pharmacokinetic Modeling of Missed Mycophenolate Mofetil Doses: Impact on Exposure and Exploration of Mitigation Strategies

PONE-D-25-36180R1

Dear Dr. Caroline Monchaud,

We’re pleased to inform you that your manuscript has been judged scientifically suitable for publication and will be formally accepted for publication once it meets all outstanding technical requirements.

Kind regards,

Paulo Caceres Guido

Academic Editor

PLOS ONE

---

## [Editor Report · Acceptance letter]

PONE-D-25-36180R1

PLOS ONE

Dear Dr. Monchaud,

I'm pleased to inform you that your manuscript has been deemed suitable for publication in PLOS ONE. Congratulations! Your manuscript is now being handed over to our production team.

Kind regards,

on behalf of

Dr. Paulo Caceres Guido

Academic Editor

PLOS ONE